# Occurrence of Antimicrobial Resistance in Canine and Feline Bacterial Pathogens in Germany under the Impact of the TÄHAV Amendment in 2018

**DOI:** 10.3390/antibiotics12071193

**Published:** 2023-07-15

**Authors:** Marianne Moerer, Antina Lübke-Becker, Astrid Bethe, Roswitha Merle, Wolfgang Bäumer

**Affiliations:** 1Institute of Pharmacology and Toxicology, Department of Veterinary Medicine, Freie Universität Berlin, Koserstraße 20, 14195 Berlin, Germany; moerer_marianne@t-online.de; 2Institute of Microbiology and Epizootics, Department of Veterinary Medicine, Freie Universität Berlin, Robert-von-Ostertag-Str. 7, Building 35, 14163 Berlin, Germany; antina.luebke-becker@fu-berlin.de (A.L.-B.); astrid.bethe@fu-berlin.de (A.B.); 3Institute for Veterinary Epidemiology and Biostatistics, Department of Veterinary Medicine, Freie Universität Berlin, Königsweg 67, 14163 Berlin, Germany; roswitha.merle@fu-berlin.de

**Keywords:** dog, cat, companion animals, legislation, antimicrobial susceptibility testing (AST), cumulative antimicrobial susceptibility

## Abstract

The occurrence of antimicrobial resistance due to the use of antimicrobials is considered to be a main cause for treatment failure of bacterial infections in humans and animals. The right of German veterinarians to use and prescribe medications such as antimicrobials is regulated by the Regulation of Veterinary Pharmacies (TÄHAV). The aim of this study was to investigate the impact of the second amendment to the TÄHAV in 2018 on the occurrence of antimicrobial resistance in selected bacterial pathogens isolated from dogs and cats in Germany. For this purpose, we analyzed antimicrobial susceptibility data from 38 German small animal practices gathered between 2015 and 2021 in cooperation with Laboklin (Labor für klinische Diagnostik GmbH & Co.KG, Bad Kissingen, Germany). Annual cumulative susceptibility data of eight bacterial species were analyzed and compared. The mean value of resistant isolates was determined for each year and supplemented by 95% confidence intervals. Encouraged by the amendment, an increase in sample submissions was observed in Germany. The highest resistance rates to the analyzed substances penicillin G, ampicillin, amoxicillin-clavulanic acid, cefovecin, and enrofloxacin were found for *Staphylococcus pseudintermedius* (*S. pseudintermedius*), *S. aureus*, and *Escherichia coli (E. coli)*. In contrast, resistance rates were low for *Pasteurella multocida* (*P. multocida*) and *β-hemolytic* streptococci. Significant resistance trends (*p* < 0.05) assumed as influenced by the TÄHAV amendment could be the significant decreases in resistance rates of *S. pseudintermedius* against penicillin G to 67% (n = 322/479), and ampicillin to 63% (n = 286/453), as well as *S. felis* against amoxicillin-clavulanic acid and cefovecin to 2% (n = 2/109), furthermore, the reduction in the occurrence of resistance of *S. aureus* against enrofloxacin to 4% (n = 3/76) in 2021. Moreover, for all species, the efficacy against the analyzed substances was maintained over the study period.

## 1. Introduction

Antimicrobials are an essential part of the successful treatment of bacterial infections [1], and the loss of their efficacy due to antimicrobial resistance is currently one of the greatest global threats to public health [2,3]. Any use of antimicrobials creates a selective pressure resulting in the emergence of resistance [2,4]. The foundation for containing the spread of resistance is to optimize the use of antimicrobials [2,4,5,6]. In line with the One Health approach, the use of antimicrobials in veterinary medicine is also a relevant factor in the global development of resistance. This is particularly relevant because antimicrobials used to treat companion animals are almost identical to those used in human medicine [7] and the intensive physical contact between pet and owner enhances the possibility of transmission of drug resistant microorganisms [8,9,10,11]. In this context, 3rd/4th generation cephalosporins and fluoroquinolones have been classified as highest priority critically important antimicrobials (HPCIA) to human health, as their efficiency is without alternative in the treatment of certain human diseases [4,12]. To reduce the use of HPCIA, it is particularly indicated to use the support of bacterial diagnostics and antimicrobial susceptibility testing (AST) to guide the selection of appropriate antimicrobials for therapy [2,4,6,13].

In Germany, the permission to dispense and prescribe medications is regulated in the Regulation on Veterinary Pharmacies (Verordnung über Tierärztliche Hausapotheken (TÄHAV)). This regulation was revised in 2018 with the aim to enhance the prudent use of antimicrobials in order to maintain the efficacy of antimicrobial agents. Therefore, AST, performed according to internationally recognized procedures, is mandatory for the use of 3rd/4th generation cephalosporines and fluoroquinolones (§12c, d TÄHAV). While these innovations provide veterinarians with a basis for good scientific practice, they are often perceived as limiting therapeutic options [14].

Standardized resistance data could be used to develop further methods to reduce the risk of resistance selection and to evaluate the impact of these interventions on antimicrobial resistance as well as animal health and welfare [2,7,15,16]. However, the results of AST from commercial laboratories have currently not been used for a nationwide scientific evaluation in Germany.

The aim of this study was to analyze the occurrence of antimicrobial resistance in companion animal pathogens in Germany under the impact of the TÄHAV amendment in 2018.

## 2. Results

Data from 38 practices were included. Eighteen participants were classified as small practices with one to two veterinarians employed and a daily patient load of less than 30. Of these, seven practices were located in small towns, two were in cities with populations of 20,000–100,000 (medium-sized cities), and nine were located in large cities. Large veterinary practices employed more than 3 veterinarians and treated more than 30 patients daily. These practices were also located in small towns (8), in medium-sized towns (3), and large cities (5). The number of small and large practices was evenly distributed across 11 of 16 German states (Schleswig-Holstein, Mecklenburg-Western Pomerania, Brandenburg, Berlin, Saxony, Bavaria, Baden-Württemberg, Rhineland-Palatinate, Hesse, North Rhine-Westphalia, Lower Saxony). In addition, four clinics participated in the data analysis, two of which were in medium-sized cities and two in large cities. They were located in four different German states (Bavaria, Rhineland-Palatinate, Saxony, North Rhine-Westphalia).

During the previous survey [17], 79% (n = 30/38) of the current participants stated to use HPCIA less frequently due to the TÄHAV amendment. In addition, 66% (n = 25/38) reported using penicillins generally as initial therapy due to the revision. AST were requested more frequently by 76% (n = 29/38) of participants, but 41% (n = 11/27) reported testing mostly in parallel with antibiotic therapy and only 15% (n = 4/27) of participants actually waited for the test result to select an appropriate antimicrobial agent.

In total, 1563 samples were submitted by 30 practices in 2015; 1758 samples by 36 practices in 2016; 38 practices sent in 1619 samples in 2017; 2266 samples in 2018; 2088 samples in 2019; 2097 samples in 2020; and 1979 samples in 2021. A significant increase in sample submissions (*p* = 0.01) was found in 2018. The additional six practices in 2016 accounted for 71 isolates tested and the additional two practices in 2017 led to 25 isolates tested. Thus, the additional practices did not result in a significant increase in investigated isolates. The number of sample submissions per year is shown in Figure 1.

The number of sample submissions without bacterial growth was not significantly affected by the TÄHAV amendment in 2018. Samples without bacterial growth were predominantly from urine (23%, n = 618/2691) and ears (18%, n = 473/2691), besides samples of unknown origin (29%, n = 793/2691).

A total of 19,792 isolates were identified. Among them, 3790 (19.1%) isolates were classified as contamination or microbiota. Isolates from canine specimens (n = 14,881/19,792; 75.2%) were predominantly from ears (n = 3058; 20.5%), wounds (n = 1332; 9.0%), skin (n = 1045; 7.0%), and urine (n = 1167; 7.8%). Isolates from cats (n = 5111/19,792, 25.8%) were mainly from the nasopharynx (n = 932; 18.2%), wounds (n = 577; 11.3%), or urine (n = 706; 13.8%). A large proportion of isolates from material of unknown origin was found in both dogs and cats (n = 4442; 30.3% and n = 1625; 31.8%, respectively).

From canine sample submissions (n = 9651 samples, n = 14,881 isolates), *Staphylococcus pseudintermedius (S. pseudintermedius)* (n = 3012; 20.5%), *Escherichia coli (E. coli)* (n = 1790; 12.2%), and *β-hemolytic* streptococci (n = 1086; 7.4%) were identified most frequently. *Pasteurella multocida (P. multocida)* (n = 790; 15.5%), *Staphylococcus felis (S. felis)* (n = 661; 12.9%), and *E. coli* (n = 585; 11.4%) were most frequently isolated from feline samples (n = 3719 samples, n = 5111 isolates). The total number of pathogens analyzed per year and the proportion of isolates per animal species are shown in Table 1.

### 2.1. Antimicrobial Resistance of Staphylococcus pseudintermedius

Both prior to and post amendment of the TÄHAV in 2018, the highest resistance rates were observed for penicillin G and ampicillin. In 2015 and 2016, the percentage of resistant isolates was significantly higher (*p* < 0.05) at (78%; n = 286/366 and n = 329/422, respectively) than in subsequent years (2017 72%, n = 296/409). Another significant (*p* < 0.05) decrease in the percentage of *S. pseudintermedius* isolates evaluated as resistant since 2017 occurred in 2021 (67.2%, n = 322/479). Resistance to ampicillin varied between 72% (2015, n = 256/355) and 68% (2016, n = 275/407) in the years 2015 to 2020 with a significant reduction in resistance since 2015 in 2021 (63.1%, n = 286/453). A lower proportion of isolates were found to be resistant to amoxicillin-clavulanic acid and cefovecin compared to the previously described substances. Resistance rates to both agents varied from 10% to 16% and showed no significant change in trend over the years. Against enrofloxacin, resistance differed between 6% (2020, n = 26/452) and 13% (2017, n = 53/409). A significant reduction in resistance was detected from 2015 (12%, n = 45/366) to 2019 (7%, n = 35/495) (Figure 2).

### 2.2. Antimicrobial Resistance of Staphylococcus felis

The highest resistance rates were observed for penicillin G and ampicillin. At the beginning of the study, 72% (n = 34/47) of isolates were evaluated as resistant to penicillin G and 59% (n = 27/47) as resistant to ampicillin. A significant (*p* < 0.05) decrease in resistance was observed for both agents until 2017 (39%, n = 25/65). A slight non-significant decrease in resistance was also observed in the following years (2020 30%, n = 38/127). Lower rates of resistance were identified to amoxicillin-clavulanic acid as well as cefovecin. While in 2015, 23% (n = 11/47) and 28% (n = 13/47) of isolates were resistant against amoxicillin-clavulanic acid and cefovecin, a significant reduction (*p* < 0.05) of resistant isolates was found for both agents in 2016 (8%, n = 5/65 and 6%, n = 4/65, respectively). A further significant decrease (*p* < 0.05) in resistance for those two agents was found in 2021 (2%, n = 2/109). A similar finding was shown for resistance to enrofloxacin. A significant decrease (*p* < 0.05) in resistant *S. felis* isolates since 2015 was first observed in 2016 (3%, n = 2/65) and since then again in 2019 (1.5%, 2/131) (Figure 3).

### 2.3. Antimicrobial Resistance of Staphylococcus aureus

*S. aureus* isolates showed highest resistance rates to penicillin G and ampicillin. While in 2015 88% (n = 52/59) of isolates were resistant to penicillin G, a significant reduction in resistance (*p* < 0.05) occurred in 2017 (63%, n = 31/49). Since 2017 the proportion of resistant isolates increased significantly (*p* < 0.05) to 76% (n = 58/76) in 2021. Resistance to ampicillin was assessed separately for canine and feline isolates since different breakpoints were used for the evaluation [18]. Resistance to ampicillin ranged from 65% (2017, n = 20/31) to 83% (2018, n = 30/36) in canine isolates and from 90% (2016, n = 19/21) to 59% (2018, n = 23/39) in feline isolates without any significant trend changes. The evaluation of resistance rates against both amoxicillin-clavulanic acid and cefovecin showed a significant decrease (*p* < 0.05) from 2016 (49%, n = 30/62) to 2017 (9%, n = 4/49) with a directly following increase in resistance rates in 2018 (20%, n = 15/75) and again a slight reduction in 2021 (12%, n = 9/76). Similarly, resistance rates to enrofloxacin decreased significantly (*p* < 0.05) in 2017 (10%, n = 5/49) with a further significant decrease (*p* < 0.05) in 2021 (4%, n = 3/76) (Figure 4).

### 2.4. Antimicrobial Resistance of Canine E. coli Isolates from the Urinary Tract (UTI)

The highest resistance rates were found against ampicillin. The percentage of resistant bacteria varied between 39% (2017, n = 18/46) and 20% (2019, n = 12/59). No significant trend change in the occurance of resistance was observed during the study period. Likewise, no significant changes in resistance rates were identified to amoxicillin-clavulanic acid. Resistance rates ranged from 2% (2016, n = 1/44) to 11% (2017, n = 5/46). While a significant increase (*p* < 0.05) in cefovecin resistant isolates was observed in 2016 (23%, n = 10/44) a significant decline (*p* < 0.05) in resistance rates was found until 2019 (5%, n = 3/59), further declining in 2021 (3%, n = 1/34). Against enrofloxacin, the proportion of resistant isolates decreased from 22% (2015, n = 8/37) to 6% (2021, n = 2/34) with a significant decline (*p* < 0.05) in 2019 (7%, n = 4/59) (Figure 5).

### 2.5. Antimicrobial Resistance of Escherichia coli Isolates from Skin and Soft Tissue (SST)

Only the occurrence of resistance against cefovecin and enrofloxacin was investigated, as, according to CLSI, isolates of *Enterobacterales* from body sites other than urine should not be reported as susceptible to aminopenicillins or amoxicillin-clavulanic acid due to low tissue concentrations achievable. The evaluation of cefovecin resistant isolates revealed significant (*p* < 0.05) fluctuations in direct consecutive years. However, since 2015 (28%, n = 50/179), a significantly decreasing trend (*p* < 0.05) in the occurrence of resistant *E. coli* was observed until 2021 (11%, n = 35/305). In 2015, 17% (n = 30/179) of isolates were resistant to enrofloxacin, and in 2018, a significant (*p* < 0.05) decrease in resistance was observed (8%, n = 27/349) (Figure 6).

### 2.6. Antimicrobial Resistance of Proteus mirabilis

Cefovecin resistance rates of *P. mirabilis* showed significant fluctuations in directly consecutive years. However, since 2015 (24%, n = 12/51), a significant decrease (*p* < 0.05) in the occurrence of resistance was found until 2020 (13%, n = 9/67). No significant changes in the occurrence of resistance to enrofloxacin were seen. Thereby, the proportion of resistant isolates ranged from 15% (2017, n = 8/53) to 6% (2021, n = 4/73) (Figure 7).

### 2.7. Antimicrobial Resistance of Klebsiella spp.

*Klebsiella pneumoniae* (60.7%), *Klebsiella oxytoca* (24.4%), *Klebsiella variicola* (9.6%), and *Klebsiella aerogenes* (5.3%) were grouped as *Klebsiella* spp. because their evaluation was performed according to the same breakpoints.

Resistance to cefovecin declined significantly (*p* < 0.05) from 2016 (32%, n = 7/22) to 2018 (10%, n = 4/40), resistance ranged further between 10% (2020, n = 4/41) and 21% (2021, n = 6/29). The proportion of enrofloxacin resistant isolates varied between 11% (2017, n = 2/19) and 0% (2018, n = 0/40). The years 2018 and 2019 showed significantly less resistance (*p* < 0.05) compared to the other study years (Figure 8).

### 2.8. Antimicrobial Resistance of Pasteurella multocida

Less than 7% of the isolates were resistant to one or more of the investigated antimicrobial agents.

Against cefovecin, a significant change (*p* < 0.05) in resistance rates was observed. While 6.5% (n = 12/188) of isolates were resistant in 2018, resistance rates decreased significantly (*p* < 0.05) to 1% (n = 2/189) from 2019 onwards (Figure 9).

### 2.9. Antimicrobial Resistance of β-Hemolytic Streptococcus spp.

No resistance to β-lactam antimicrobials was detected in this study. Only in 2020, 2 of 202 isolates showed resistance to enrofloxacin. Thus, no significant change in the occurrence of resistance was detected for the tested agents.

## 3. Discussion

The World Health Organization compiled a list of critically important agents whose efficacy must be maintained for the treatment of human diseases [19]. The use of these agents is being restricted by an increasing number of guidelines, including the second amendment to the TÄHAV in Germany in 2018, in order to optimize the use of these antimicrobials to maintain their effectiveness [20]. From the HPCIA group, fluoroquinolones and 3rd generation cephalosporins were mainly used in small animal medicine due to their good efficacy and ease of use [14,21,22]. According to studies, their use was less than 10% of the applied antibiotics in Germany and the TÄHAV amendment was described as a reason for reduced use [17,23]. The most commonly applied agents in small animal medicine belong to the group of aminopenicillins [17,23,24], which, due to their broad spectrum of activity, are increasingly used as an alternative medication to HPCIA since the amendment of the TÄHAV, also to avoid AST [17]. Even if an influence of antibiotic application quantities on the development of resistance rates is assumed [25], this factor can only be reliably included into analyses from 2026 onwards with the legally required reporting of antibiotic consumption quantities.

As of 2018, a significant increase in sample submissions was observed in Germany. Therefore, it may be concluded that the amendment had an impact on the testing behavior of the participating veterinarians [17], assuming that the number of patients examined in the participating practices remained steady over the analyzed time period. According to previous surveys, the patient owner’s willingness and ability to pay for AST influenced the treatment decision greatly [26,27]. By implementing the antibiogram requirement for the use of fluoroquinolones and 3rd/4th generation cephalosporines, veterinarians were provided with a rationale for good veterinary practice [14]. Especially among the most frequently sent materials, a large proportion of samples was found from which no bacteria could be isolated and thus an antibiotic therapy could be identified as avoidable.

The number of submitted samples for microbiological examination and susceptibility testing was significantly higher for dogs than for cats. However, the proportion of dogs and cats presented for examination in our participating practices is not known. Nevertheless, according to the literature, diseases of the skin and ears require antibiotic therapy particularly often which occur more frequently in dogs than cats [4,13,17,28] and are associated with increasingly resistant bacteria [29,30,31].

### 3.1. Staphylococcus spp.

Staphylococci are opportunistic pathogens bacteria that form part of the natural microbiome of skin and mucosa in animals [10,32,33]. The literature recommends aminopenicillins as first-line therapy for the treatment of infections caused by staphylococci [1,34]. However, resistance to penicillin G and ampicillin was found to be most common among *Staphylococcus* spp. as also described in the literature [31,32,35]. According to CLSI, an isolate resistant to penicillin G can be considered resistant to all penicillinase labile agents, such as aminopenicillins [36]. In contrast to M100, there are separate species-specific breakpoints for ampicillin used for evaluation in Vet01, which may have led to inconsistent interpretations in 133 *Staphylococcus* isolates during the analysis from 2015 to 2021.

Current resistance rates of *Staphylococcus* spp. determined during this study are comparable to values reported in the literature [31,35].

However, in contrast to the data of the GERM-Vet study, *S. aureus* and *S. felis* isolates were increasingly resistant against various antimicrobial agents in 2015 and 2016, respectively [37]. These deviations could be related to changes in internal laboratory processes that can no longer be tracked. However, an increase in the emergence of antimicrobial resistance could not be detected for the investigated active substances. Therefore, the efficacy of these antimicrobials has been maintained since the amendment of the TÄHAV; the GERM-Vet study also confirms these results and even supports the tendency of a decreasing occurrence of resistance [37]. Moreover, decreasing resistance to enrofloxacin was observed for all *Staphylococcus* spp. during this study. While the proportion of resistant *S. aureus* isolates only decreased significantly in 2021 compared to previous years, this trend can already be observed for *S. felis* as early as 2018 and even for *S. pseudintermedius* from 2017 onwards. As well, the Bundesamt für Verbraucherschutz und Lebensmittelsicherheit (BVL) found a significant reduction in resistance of *S. aureus* to enrofloxacin from 32% in 2012 to 11% in 2020. In contrast to our data, the GERM-Vet study described increasing levels of enrofloxacin resistance in *S. pseudintermedius* from 6% (2018) to 11% in 2020 in dogs pretreated with antimicrobials. Previous studies found tendencies that antibiotic pretreatment of animals could have an effect on the increased incidence of resistant isolates [31,37]. Information on the pretreatment of the animals was not available in the current study. These results could also be biased by different testing behavior of veterinarians as a result of the TÄHAV amendment as indicated by the rising number of samples submitted per year [17].

### 3.2. Enterobacterales

Enterobacteria are Gram-negative, facultative anaerobic rod-shaped bacteria. They are part of the natural microbiome of the intestinal tract of humans and animals [38]. However, in addition to physiological colonization, some strains cause severe intestinal and extraintestinal diseases [32,39].

Due to their natural occurrence in large numbers, members of this family with potentially zoonotic properties represent a major reservoir of resistance genes that may be transferred horizontally and are of public health concern [29,40,41,42].

While aminopenicillins do not reach concentrations high enough for effective therapy in the tissue [18], these agents are excreted renally unchanged and are highly concentrated in urine, allowing its application for therapy of lower urinary tract infections caused by *E. coli* [43]. Therefore, a separate evaluation of isolates from infections of the urinary tract was performed.

The proportion of resistant isolates against HPCIA is equally low in all indications. This was supported by the results of the GERM-Vet study [37], while another study examining samples from antimicrobial pretreated animals described significantly higher resistance rates in *E. coli* [44]. The observed trend of declining resistance rates against enrofloxacin for both SST and UTI isolates was also confirmed for canine UTI *E. coli* in the GERM-Vet study data from 2012 to 2019 [37] and could reflect a more conscious use of antimicrobials in the treatment of cystitis [14].

Although the data from the current study found decreasing resistance rates against cefovecin for all species tested, it is important to be aware when evaluating these results that UTI breakpoints were also applied to SST isolates by the investigating laboratory, however, the accumulation in soft tissue is distinctly different from the concentration in urine. Furthermore, as of 2018, the CLSI provided an adjusted breakpoint for cefovecin UTI, but the time of implementation by the laboratory is unknown.

For the comparison of resistance in *Klebsiella* spp. and *P. mirabilis* in dogs and cats, the study of the Bundesverband für Tiergesundheit (BfT) from 2007 was used, as susceptibility testing was performed using the same method as in the current study. However, the BfT study described a much higher proportion of enrofloxacin resistant pathogens with 27% of *P. mirabilis* and 29% of *Klebsiella* spp. being resistant [45]. Only UTI isolates were examined for the BfT study, whereas in the current study isolates of various origins were analyzed. Additionally, Harada et al. found significant differences in resistance between a variety of indications for *P. mirabilis* [41].

Even though the generally observed decreasing incidence of resistance as of 2018 could be associated with the strongly increased number of Enterobacterales isolates tested per year since the TÄHAV amendment, due to changes in testing behavior of the participating veterinarians, the GERM-Vet study supports the finding of maintenance of antimicrobial efficiency [37].

### 3.3. Pasteurella multocida

*P. multocida* are Gram-negative, rod-shaped bacteria that are part of the natural microbiome of the mucosa of the upper respiratory tract of cats but also dogs [46,47].

Infections caused by *P. multocida* are especially common in cats with bite wound infections or infections of the respiratory tract [46]. Literature recommends antibiotic treatment with aminopenicillins [47]. Data from this study as well as further studies showed good susceptibility of those agents used for an initial therapy [37,48]. However, the use of Convenia^®^ (Zoetis, Berlin, Germany), a 3rd generation cephalosporin with depot effect, was described in several studies due to the application advantages in cats [4,17,21,22]. The TÄHAV amendment may have led to more frequent bacteriological testing of bite wound infections and respiratory diseases, to be allowed to apply Convenia^®^. Those cases would have been successfully treated without AST prior to the legal obligation. Therefore, an increase of susceptible isolates in the data could have occurred and led to the significant decrease in resistance observed during this study.

### 3.4. β-Hemolytic Streptococcus spp.

Streptococci are opportunistic pathogens of a wide variety of hosts, causing different infectious diseases, especially otitis and pyoderma [49,50,51]. While *β-hemolytic streptococci* are among the most commonly isolated pathogens in dogs, they are only exceptionally associated with resistance as found in the current data as well as further studies [30,52].

### 3.5. Limitations

It can be assumed that veterinarians who participated in this study and agreed to share the results of AST have a greater interest and awareness of the antimicrobial resistance problem, which may have resulted in data not reflecting the actual resistance situation in Germany. Inclusion of veterinary practices from all over Germany was done to avoid bias in resistance data due to geographic or practice-specific influences. In order to make the data more comparable and standardized, we decided to work with only one laboratory. Since retrospective data were evaluated, practice internal processes such as the decision to collect samples, the sample collection procedure, and sample processing, as well as information on pretreatment of patients are not available. It cannot be ruled out that these factors may have led to bias in the results. Especially since the number of submitted samples significantly increased as of the TÄHAV amendment, but the reasons for this are unknown. More frequent testing of uncomplicated infections since the TÄHAV amendment might have enhanced the number of susceptible isolates in the data pool and thus led to the significant decrease in resistance observed in this study. Although the results of our participants’ survey suggest more prudent use of antibiotics, with HPCIAs used less frequently and penicillins deliberately selected as alternative antimicrobials, the first significant changes in the occurrence of resistance due to legal adjustments as the TÄHAV amendment are expected no earlier than three to five years after implementation [53]. Therefore, only significant trends in resistance as of 2021 could be assumed as influenced by the TÄHAV amendment such as the significant decreases in resistance rates of *S. pseudintermedius* against penicillin G and ampicillin, as well as the reduction in the occurrence of resistance of *S. aureus* against enrofloxacin.

Furthermore, in addition to AST results, the general health and condition of the immune system is also crucial when treating animals. AST should therefore only be considered as a supportive tool for treatment decisions.

## 4. Materials and Methods

### 4.1. Recruitment

To prevent bias in the data due to sample processing and evaluation, it was decided to work with only one laboratory, Laboklin (Labor für klinische Diagnostik GmbH & Co.KG, Bad Kissingen, Germany). During a previous survey, 378 German veterinarians were asked about their use of antimicrobials [17]. Forty-one of those veterinary practices collaborated with Laboklin for bacteriological examination and AST and provided their contact information after completing the questionnaire. All 41 practices gave their consent to provide susceptibility data for analysis. For three practices, no AST data were available from the period before the TÄHAV amendment in 2018, therefore, they were excluded from the analysis.

### 4.2. Bacterial Identification and Antimicrobial Susceptibility Testing (AST)

The bacteriological examination was performed by the commercial laboratory Laboklin. Bacterial identification was performed via MALDI TOF (Microflex LT and Biotyper sirius one, Bruker Daltonik GmbH, Bremen, Germany). For susceptibility testing, the minimum inhibitory concentration (MIC) was determined by broth microdilution using microtiter plates with individual customized layouts (MERLIN Micronaut System, Bornheim-Hersel, Germany). The microtiter plates were read photometrically. Testing and evaluation of the results was performed according to internationally recognized standardized procedures of the Clinical and Laboratory Standards Institute (CLSI) M100, Vet01, and Vet08 [18,36]. The results of bacterial identification and qualitative classification of the MIC values, together with information on the sample type and patient species were provided by the laboratory.

### 4.3. Coding

The qualitative classification of isolates was provided in Microsoft Excel^®^ 2018 (Microsoft Corporation, 2018, Microsoft Excel. Retrieved from https://office.microsoft.com/excel, accessed on 30 June 2021) tables for 2015 to 2021. In Excel, recoding took place, summarizing the laboratory-provided names of the submitted samples according to synonyms and content-similar keywords.

Some bacterial species have been grouped together under their genus name; therefore *Klebsiella* spp. includes *K. aerogenes*, *K. oxytoca*, *K. pneumoniae*, and *K. variicola.*

### 4.4. Descriptive Analysis

Data obtained by the laboratory included the interpretation (sensitive (S)—intermediate (I)—resistant (R)) of the MIC. The analysis in this study was limited to relevant antimicrobials, particularly those whose efficiency may have been influenced by the TÄHAV amendment [17] and relevant pathogen-agent indication. The CLSI is the only organization that currently provides animal species-specific clinical breakpoints for some pathogen–drug combinations [18]. Nevertheless, for certain antimicrobial agents, no species-specific clinical breakpoints are available [54]. In the absence of veterinary-specific clinical breakpoints for isolates from dogs and cats, human-specific clinical breakpoints from CLSI document M100 were applied.

### 4.5. Statistical Analysis

IBM SPSS Statistics version 29 (IBM Corp. Released 2022. IBM SPSS Statistics for Windows, version 29.0. IBM Corp.: Armonk, NY, USA) was used for statistical analysis.

For metric description, a sensitive result was assigned the numeral one, intermediate results the numeral two, and resistant results were represented by a three. Through these numerical assignments, an annual mean value representing the resistance rate was calculated. The development of the mean values was determined for each pathogen-agent combination in a line graph. Since the data are only a sample, the 95% confidence interval (CI) was calculated around each mean value. The mean value of 2015, the first year of this study, was set as the reference. If the mean values and their CIs of the following years were completely outside this reference line, a significant change in the resistance rate was assumed since *p*-values were less than 0.05 in these cases. The mean value of the year with the significant shift was set as reference for the assessment of changes in resistance of further years. All data were analyzed by descriptive analysis. The results are shown in the Appendix A.

## 5. Conclusions

Veterinarians in Germany test pathogens more frequently for susceptibility as of 2018, which may lead to the assumption that the amendment of the TÄHAV has encouraged a more prudent use of antimicrobials. For all the bacterial species, the efficacy of analyzed substances was maintained over the study period. Nevertheless, the development and spread of antibiotic resistance still poses a public health risk and therefore requires further monitoring. Future research could focus on making commercially produced AST usable. In combination with the already legally required reporting of antibiotic consumption levels in dogs and cats from 2026, valuable information could be collected for the evaluation of legal measures.

## Figures and Tables

**Figure 1 antibiotics-12-01193-f001:**
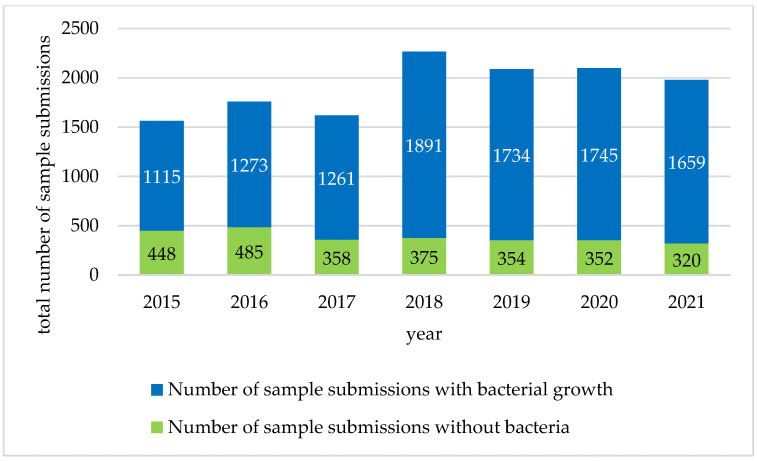
Number of sample submissions. The total number of samples for antimicrobial susceptibility testing in the years 2015 to 2021. The number of samples without bacterial growth each year is shown separately.

**Figure 2 antibiotics-12-01193-f002:**
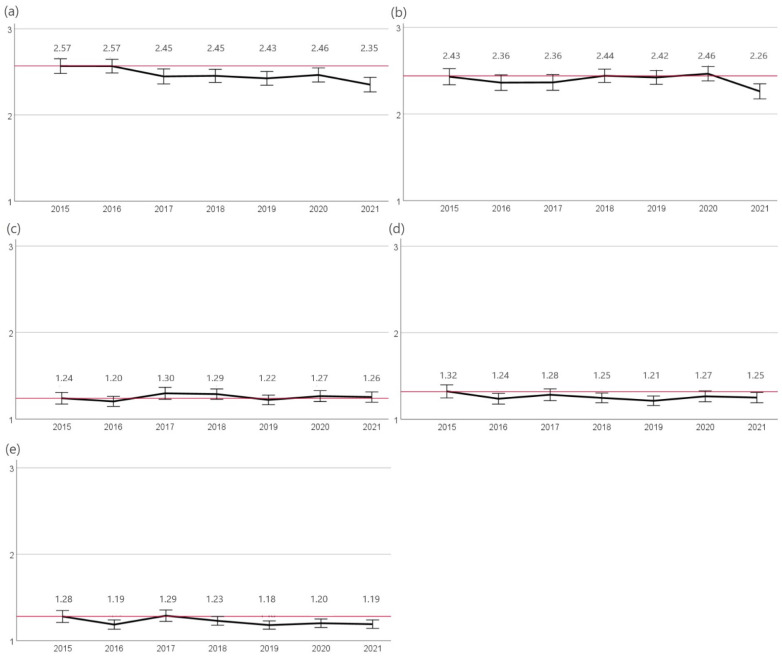
Resistance of *S. pseudintermedius***.** The figure shows the mean values of *S. pseudintermedius* isolates calculated from resistant = 3, intermediate = 2, and sensitive = 1 of the agents (**a**) penicillin G, (**b**) ampicillin, (**c**) amoxicillin-clavulanic acid, (**d**) cefovecin, and (**e**) enrofloxacin for the years 2015 to 2021. Around each mean value, the 95% CI was shown. The reference line to the mean of the first year of the study was shown in red. The number of isolates were 2015: n = 366, 2016: n = 422, 2017: n = 409, 2018: n = 533, 2019: n = 495, 2020: n = 452, and 2021: n = 479.

**Figure 3 antibiotics-12-01193-f003:**
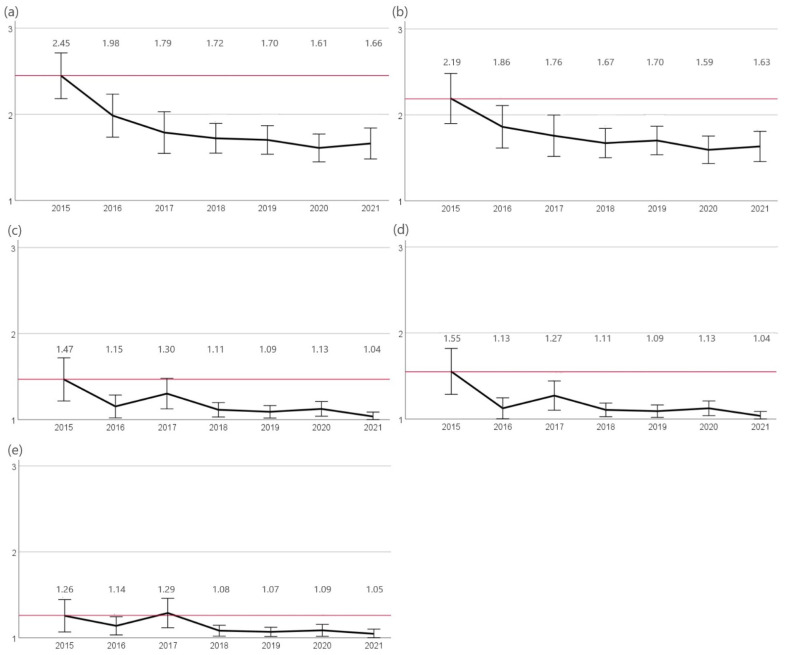
Resistance of *S. felis*. The figure shows the mean values of *S. felis* isolates calculated from resistant = 3, intermediate = 2, and sensitive = 1 of the agents (**a**) penicillin G, (**b**) ampicillin, (**c**) amoxicillin-clavulanic acid, (**d**) cefovecin, and (**e**) enrofloxacin for the years 2015 to 2021. Around each mean value, the 95% CI was shown. The reference line to the mean of the first year of the study was shown in red. The number of isolates were 2015: n = 47, 2016: n = 65, 2017: n = 66, 2018: n = 122, 2019: n = 131, 2020: n = 128, and 2021: n = 109.

**Figure 4 antibiotics-12-01193-f004:**
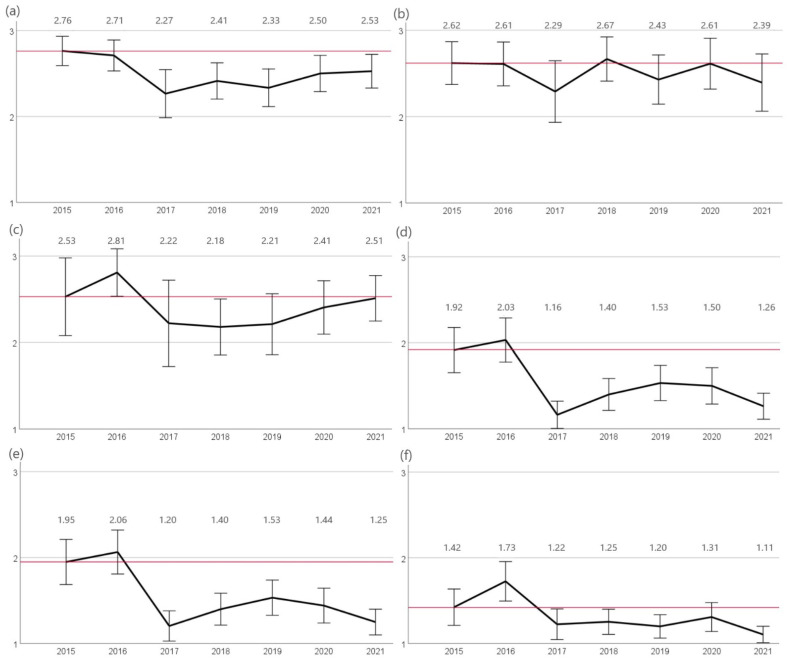
Resistance of *S. aureus*. The figure shows the mean values of *S. aureus* calculated from resistant = 3, intermediate = 2, and sensitive = 1 of the agents (**a**) penicillin G, (**b**) ampicillin canine, (**c**) ampicillin feline, (**d**) amoxicillin-clavulanic acid, (**e**) cefovecin, and (**f**) enrofloxacin for the years 2015 to 2021. Around each mean value, the 95% CI was shown. The reference line to the mean of the first year of the study was shown in red. The number of isolates were 2015: n = 59 (canine n = 42, feline n = 17), 2016: n = 62 (canine n = 41, feline n = 21), 2017: n = 49 (canine n = 31, feline n = 18), 2018: n = 75 (canine n = 36, feline n = 39), 2019: n = 75 (canine n = 42, feline n = 33), 2020: n = 68 (canine n = 31, feline n = 37), and 2021: n = 76 (canine n = 33, feline n = 43).

**Figure 5 antibiotics-12-01193-f005:**
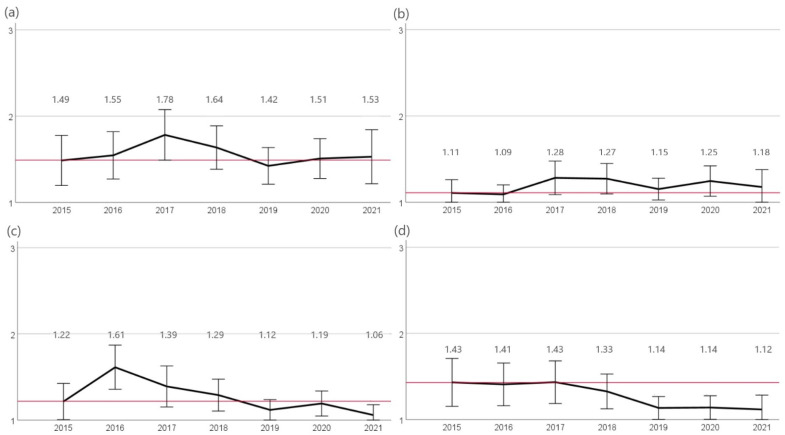
Resistance of *E. coli* UTI. The figure shows the mean values of *E. coli* UTI isolates calculated from resistant = 3, intermediate = 2, and sensitive = 1 of the agents (**a**) ampicillin, (**b**) amoxicillin-clavulanic acid, (**c**) cefovecin, and (**d**) enrofloxacin for the years 2015 to 2021. Around each mean value, the 95% CI was shown. The reference line to the mean of the first year of the study was shown in red. The number of assessed isolates were 2015: n = 37, 2016: n = 44, 2017: n = 46, 2018: n = 55, 2019: n = 59, 2020: n = 57, and 2021: n = 34.

**Figure 6 antibiotics-12-01193-f006:**
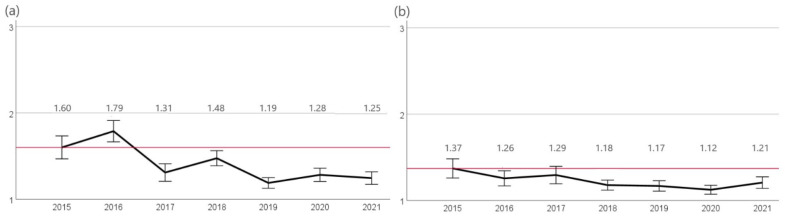
Resistance of *E. coli* SST. The figure shows the mean values of *E. coli* SST calculated from resistant = 3, intermediate = 2, and sensitive = 1 of the agents (**a**) cefovecin and (**b**) enrofloxacin for the years 2015 to 2021. Around each mean value, the 95% CI was shown. The reference line to the mean of the first year of the study was shown in red. The number of assessed isolates were 2015: n = 179, 2016: n = 219, 2017: n = 187, 2018: n = 349, 2019: n = 316, 2020: n = 307, and 2021: n = 305.

**Figure 7 antibiotics-12-01193-f007:**
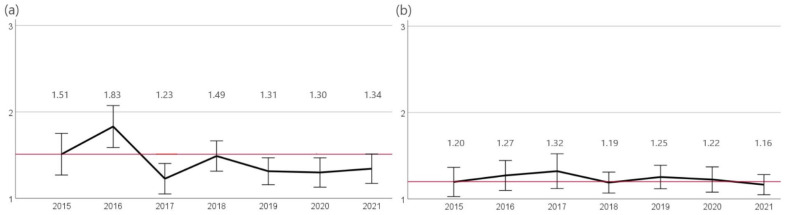
Resistance of *P. mirabilis*. The figure shows the mean values of *P. mirabilis* isolates calculated from resistant = 3, intermediate = 2, and sensitive = 1 of the agents (**a**) cefovecin and (**b**) enrofloxacin for the years 2015 to 2021. Around each mean value, the 95% CI was shown. The reference line to the mean of the first year of the study was shown in red. The number of assessed isolates were 2015: n = 51, 2016: n = 60, 2017: n = 53, 2018: n = 90, 2019: n = 83, 2020: n = 67, and 2021: n = 73.

**Figure 8 antibiotics-12-01193-f008:**
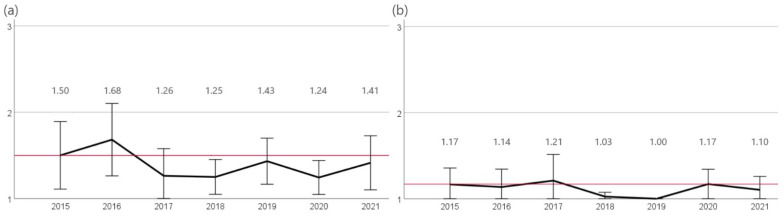
Resistance of *Klebsiella* spp. The figure shows the mean values of *Klebisiella* spp. calculated from resistant = 3, intermediate = 2, and sensitive = 1 of the agents (**a**) cefovecin and (**b**) enrofloxacin for the years 2015 to 2021. Around each mean value, the 95% CI was shown. The reference line to the mean of the first year of the study was shown in red. The number of assessed isolates were 2015: n = 18, 2016: n = 22, 2017: n = 19, 2018: n = 40, 2019: n = 37, 2020: n = 41, and 2021: n = 29.

**Figure 9 antibiotics-12-01193-f009:**
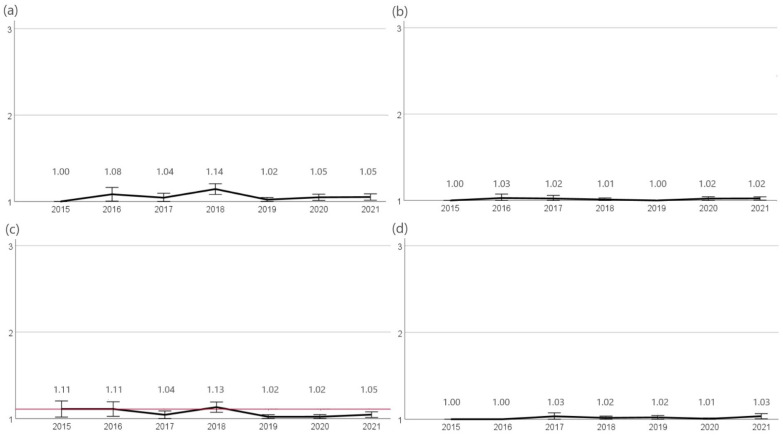
Resistance of *P. multocida*. The figure shows the mean values of *P. multocida* isolates calculated from resistant = 3, intermediate = 2, and sensitive = 1 of the agents (**a**) ampicillin, (**b**) amoxicillin-clavulanic acid, (**c**) cefovecin, and (**d**) enrofloxacin for the years 2015 to 2021. Around each mean value, the 95% CI was shown. The reference line to the mean of the first year of the study was shown in red. The number of assessed isolates were 2015: n = 63, 2016: n = 73, 2017: n = 91, 2018: n = 188, 2019: n = 189, 2020: n = 186, and 2021: n = 174.

**Table 1 antibiotics-12-01193-t001:** Number of isolates. Overview of the total number of isolates per bacterial species, as well as the percentages of isolates from dogs and cats for the years 2015 to 2021.

	2015		2016		2017		2018		2019		2020		2021	
	Frequency	Percent	Frequency	Percent	Frequency	Percent	Frequency	Percent	Frequency	Percent	Frequency	Percent	Frequency	Percent
*S. pseudintermedius*	366	100	422	100	409	100	533	100	495	100	452	100	479	100
dog	355	97.0	407	96.4	394	96.3	510	95.7	469	94.7	428	94.7	453	94.6
cat	11	3.0	15	3.6	15	3.7	23	4.3	26	5.3	24	5.3	26	5.4
*S. felis*	47	100	65	100	66	100	122	100	131	100	128	100	109	100
dog	1	2.1	0	0	1	1.5	3	2.5	0	0	1	0.8	1	0.9
cat	46	97.9	65	100.0	65	98.5	119	97.5	131	100.0	127	99.2	108	99.1
*S. aureus*	59	100	62	100	49	100	75	100	75	100	68	100	76	100
dog	42	71.2	41	66.1	31	63.3	36	48.0	42	56.0	31	45.6	33	43.4
cat	17	28.8	21	33.9	18	36.7	39	52.0	33	44.0	37	54.4	43	56.6
*E. coli* SST ^1^	179	100	219	100	187	100	349	100	316	100	307	100	305	100
dog	148	82.8	187	85.4	148	79.1	262	75.1	239	75.6	233	75.9	240	78.7
cat	31	17.2	32	14.6	39	20.9	87	24.9	77	24.4	74	24.1	65	21.3
*E. coli* UTI ^2^ canine	37	100	44	100	46	100	55	100	59	100	57	100	34	100
*P. mirabilis*	51	100	60	100	53	100	90	100	83	100	67	100	73	100
dog	48	94.1	57	95.0	49	92.5	82	91.1	79	95.2	60	89.6	67	91.8
cat	3	5.9	3	5.0	4	7.5	8	8.9	4	4.8	7	10.4	6	8.2
*Klebsiella* spp.	18	100	22	100	19	100	40	100	37	100	41	100	29	100
dog	17	94.4	22	100.0	16	84.2	36	90.0	33	89.2	34	82.9	26	89.7
cat	1	5.6	0	0	3	15.8	4	10.0	4	10.8	7	17.1	3	10.3
*P. multocida*	63	100	73	100	91	100	188	100	189	100	186	100	174	100
dog	11	17.5	26	35.6	28	30.8	26	13.8	27	14.3	23	12.4	33	19.0
cat	52	82.5	47	64.4	63	69.2	162	86.2	162	85.7	163	87.6	141	81.0
Streptococci *β-hemolytic*	137	100	146	100	139	100	232	100	215	100	202	100	156	100
dog	125	91.2	143	97.9	125	89.9	204	87.9	184	85.6	174	86.1	131	84.0
cat	12	8.8	3	2.1	14	10.1	28	12.1	31	14.4	28	13.9	25	16.0

^1^ skin and soft tissue; ^2^ urinary tract.

## Data Availability

The datasets used and/or analyzed during the current study are available from the corresponding author on reasonable request.

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
