# Peer review of "Occurrence of Antimicrobial Resistance in Canine and Feline Bacterial Pathogens in Germany under the Impact of the TÄHAV Amendment in 2018"

_antibiotics, 2023, doi:10.3390/antibiotics12071193_

Round 1

Reviewer 1 Report

This manuscript is well-written; however, I have some comments and questions:

1. The idea behind the amendment of the national law is to minimize the improper use of HPCIA (fluoroquinolones and III-IV generation cephalosporins), to tackle AMR. This manuscript addresses the impact of the national law enforcement on the use of antibiograms (in terms of numbers of samples submitted for analysis) to monitor antibiotic susceptibility and to guide therapy in the small animal practices in Germany.

The revision of the national law entered in force at the beginning of 2018, and over the course of the same year a significant increase in the number of samples submitted for analysis has been detected.

Small animal practices have been selected and included in this study on the basis of the results of a previous survey (Mörer et al, 2022) and a potential sampling bias has been openly acknowledged, which is correct, because selection bias may occur in clinic-based monitoring. Apart from that, I noticed that, even if it is assumed that the number of patients examined in the participating practices remained steady over the study period, the number of practices included in the study increased from 30 (in 2015) to 38 (in 2018).

Could you please explain why an increased number of veterinarians joined the study from 2016 and discuss the potential impact of this variable on the results (resistance trends)?

2. It has been argued that the behavior of the veterinarians has been influenced by the revision of the law and that probably more frequent testing of uncomplicated infections led to an increased number of susceptible isolates. This implies that in this study the significant resistance trends have been strongly influenced by the number of samples submitted for analysis, and not by the effective adoption of good stewardship practices (including the use of the most appropriate drug after testing). In fact, data about the diagnostic decision-making (i.e. number of tests) are available, but there is no evidence about the following therapeutic decision-making, which plays a crucial role in the development of resistance.

In the conclusions it has also been pointed out that the amendment has encouraged a more prudent use of antimicrobials. I totally agree that it is crucial to highlight the impact of the law enforcement on antimicrobial use after testing, however, could you please provide more evidence about it? For instance, even if the study is retrospective and in Germany there is no mandatory traceability system yet, would it be possible to access the vet records, to obtain information about the prescription of antimicrobials? This could give more insights about the potential reduction of the prescription of HPCIA after testing implementation and, as a results, about the impact on the decrease in resistance rates and about the possibility to minimize the risk of AMR.

3. If clients deny the permission for specimens sampling, they cannot be forced. Consequently, veterinarians will not be allowed to proceed according to the law and maybe the number of samples will drop. Could you please discuss about the role of clients on decision-making process and on the potential impact on the implementation of the TÄHAVA

4. Is it possible to suppose that the number of prescriptions of other antimicrobials, like penicillins, may increase, just because AST is not required by law. 

Could you please discuss the potential impact of the revision of the law on the use of antimicrobials other than fluoroquinolones and cephalosporins. Do you expect an increased number of prescriptions of other antimicrobials and a worsening of resistance trends?

5. I agree that it will be possible to better investigate the impact of the national law enforcement on the use of antibiograms, on the appropriate prescription of antimicrobials and in turn also on the AMR in the long term (after three to five years after implementation). Obviously ”AMR poses a public health risk and requires further monitoring”, however, could you please enounce in this paper some strategies for monitoring and provide recommendations for future research, also considering that from 2026 reporting of drugs prescription will be mandatory by law at national level?

6. I would suggest adding a Figure with the number of samples submitted for testing each year, including the samples where no bacterial growth was detected (maybe using stacked vertical bar charts)

Author Response

Comments and Suggestions for Authors:

We thank both reviewers for their intensive review and valuable comments

Rev 1

This manuscript is well-written; however, I have some comments and questions:

  1. The idea behind the amendment of the national law is to minimize the improper use of HPCIA (fluoroquinolones and III-IV generation cephalosporins), to tackle AMR. This manuscript addresses the impact of the national law enforcement on the use of antibiograms (in terms of numbers of samples submitted for analysis) to monitor antibiotic susceptibility and to guide therapy in the small animal practices in Germany.

The revision of the national law entered in force at the beginning of 2018, and over the course of the same year a significant increase in the number of samples submitted for analysis has been detected.

Small animal practices have been selected and included in this study on the basis of the results of a previous survey (Mörer et al, 2022) and a potential sampling bias has been openly acknowledged, which is correct, because selection bias may occur in clinic-based monitoring. Apart from that, I noticed that, even if it is assumed that the number of patients examined in the participating practices remained steady over the study period, the number of practices included in the study increased from 30 (in 2015) to 38 (in 2018).

Could you please explain why an increased number of veterinarians joined the study from 2016 and discuss the potential impact of this variable on the results (resistance trends)?

You are addressing a very important issue, because the increase in the number of participants has a direct impact on the number of sample submissions. Thus, the statistics could be biased to the effect that significantly more sample submissions were detected in 2018 than in previous years. However, it was noted that the additional six practices in 2016 accounted for 71 isolates tested and the additional two practices in 2017 led to 25 isolates tested. Thus, the additional practices did not result in a significant increase in investigated isolates (ll. 93 - 96). The increase in participants can be explained by the fact that these practices became new customers of Laboklin. These are not new practice start-ups, so it was assumed that the number of patients did not change significantly over the study period and therefore did not have a significant impact on the number of sample submissions.

  1. It has been argued that the behaviour of the veterinarians has been influenced by the revision of the law and that probably more frequent testing of uncomplicated infections led to an increased number of susceptible isolates. This implies that in this study the significant resistance trends have been strongly influenced by the number of samples submitted for analysis, and not by the effective adoption of good stewardship practices (including the use of the most appropriate drug after testing). In fact, data about the diagnostic decision-making (i.e. number of tests) are available, but there is no evidence about the following therapeutic decision-making, which plays a crucial role in the development of resistance.

In the conclusions it has also been pointed out that the amendment has encouraged a more prudent use of antimicrobials. I totally agree that it is crucial to highlight the impact of the law enforcement on antimicrobial use after testing, however, could you please provide more evidence about it? For instance, even if the study is retrospective and in Germany there is no mandatory traceability system yet, would it be possible to access the vet records, to obtain information about the prescription of antimicrobials? This could give more insights about the potential reduction of the prescription of HPCIA after testing implementation and, as a results, about the impact on the decrease in resistance rates and about the possibility to minimize the risk of AMR.

We fully agree with you that objective data on the use of antimicrobials under the innovations of the TÄHAV would be a very interesting aspect. However, in the course of this study, it is not possible for us to access the practice's internal antibiotic consumption volumes. However, in the previous survey, participants answered questions about antibiotic use under the influence of the TÄHAV amendment subjectively. Assuming that responses were honestly given and subjective self-assessment matched actual use, 79% (n=30/38) of participants believed HPCIA was used less frequently due to the TÄHAV amendment. In addition, 66% (n=25/38) reported using penicillins generally as initial therapy due to the amendment. Antibiograms were requested more frequently by 76% (n=29/38) of participants, but 41% (n=11/27) reported requesting antibiograms mostly in parallel with antibiotic therapy and only 15% (n=4/27) of participants actually waited for the test result to select an appropriate antibiotic agent. (ll. 84 - 89)

  1. If clients deny the permission for specimens sampling, they cannot be forced. Consequently, veterinarians will not be allowed to proceed according to the law and maybe the number of samples will drop. Could you please discuss about the role of clients on decision-making process and on the potential impact on the implementation of the TÄHAV.

In the previous survey, we found that only half of the participants reported that their patient owners were mostly willing to pay for antibiograms (Moerer et al, 2022b). Such that the choice of antibiotic therapy is forced to be influenced by willingness and ability to pay. By implementing the antibiogram requirement for the use of HPCIA, veterinarians were provided with a rationale for good veterinary practice, which is partly accepted by patient owners according to a survey (Moerer et al, 2022a). (ll. 281 - 285)

  1. Is it possible to suppose that the number of prescriptions of other antimicrobials, like penicillins, may increase, just because AST is not required by law. 

Could you please discuss the potential impact of the revision of the law on the use of antimicrobials other than fluoroquinolones and cephalosporins. Do you expect an increased number of prescriptions of other antimicrobials and a worsening of resistance trends?

As mentioned earlier, we found in the course of the previous survey that penicillins, especially aminopenicillins, were deliberately used as alternative antimicrobial therapies. In part, this was to avoid having to order AST. Widespread use of broad-spectrum antibiotics without prior testing could theoretically lead to an increase in resistance rates, but this is not confirmed by the analysis of this study data, as well as the data from the GermVet study, which in contrast even showed a reduction in resistance of certain bacterial species to agents of the penicillin group.

  1. I agree that it will be possible to better investigate the impact of the national law enforcement on the use of antibiograms, on the appropriate prescription of antimicrobials and in turn also on the AMR in the long term (after three to five years after implementation). Obviously ”AMR poses a public health risk and requires further monitoring”, however, could you please enounce in this paper some strategies for monitoring and provide recommendations for future research, also considering that from 2026 reporting of drugs prescription will be mandatory by law at national level?

In order to better monitor the development of antibiotic resistance in the future, it would be exciting to develop a strategy to make commercially produced AST usable. In combination with the already legally required reporting of antibiotic consumption levels in dogs and cats from 2026, valuable information could be collected for the evaluation of legal measures. (ll. 474 - 477)

  1. I would suggest adding a Figure with the number of samples submitted for testing each year, including the samples where no bacterial growth was detected (maybe using stacked vertical bar charts)

Thank you very much for this good idea. The figure of examined isolates was replaced by a figure of the number of sample submissions, because the legal novelties showed a direct influence on the submission behavior of the participating veterinarians, but the number of isolates was also dependent on the number of bacterial colonies contained in the sample. It was also shown, that the number of sample submissions without bacterial growth was not significantly affected by the TÄHAV amendment 2018. Samples without bacterial growth were predominantly from urine (23%, n=618/2691) and ears (18%, n=473/2691), besides samples of unknown origin (29%, n=793/2691) (ll. 101 - 104).

Reviewer 2 Report

The manuscript entitled “Occurrence of antimicrobial resistance in canine and feline bacterial pathogens in Germany under the impact of the TÄHAV amendment in 2018” described the occurrence of antibiotic resistance in pathogenic bacteria isolated from cats and dogs during 2015-2021. In my opinion, the data from this MS is beneficial for public health policy and management or the fields of veterinary sciences.

Here are my suggestions:

Line 27-28: Staphylococcus pseudintermedius (S. pseudintermedius), Staphylococcus aureus (S. aureus), Escherichia coli (E. coli), Pasteurella multocida (P. multocida)

All bacterial names should be italicized and kindly correct all points in the whole MS.

Line 29: Streptococci should be italicized.

Lines 30 & 31: Bacterial names should be corrected.

Please provide the pros and cons of the TÄHAV amendment in the Introduction section.

Line 66-77: Please provide the name of cities and German states.

Line 87-94: Please change to Table.

Line 95-98: kindly correct the bacterial names, you can use abbreviated names.

Table 1: Klebsiella should be italicized.

Table 1: What are the means of SST and UTI? Kindly added the annotation under the Table.

Kindly check and correct the writing style of bacterial full and abbreviated names (in the whole MS).

It would be better if the authors revise all figures (Fig 2 to 9), they are not clear, and the texts are so small. The reference red line is not clear in Fig 9. Markers for significant differences by statistical analysis should be included for all figures.

Line 220: kindly correct

Line 259: Staphylococci should be italicized.

Line 269: kindly correct.

Line 375: Kindly provide the location of Laboklin (city, country)

Line 384: The type, model, and manufacturer of MALTI TOF should be provided.

Line 400: revise the name of the bacteria.

The types of antibiotics used in general practice for the treatment of bacterial infection in cats and dogs by German veterinarians should be included in the Discussion section and compared with your data. Why are they still resistant or increased susceptibility? Or do they have any limitations to using antibiotics?

What is the reason why only 5 drugs are included in this study? or they are permitted under the impact of the TÄHAV amendment.

Finally, I would suggest adding the effects of TÄHAV on veterinarians in general practice. It is hard or easy to handle or treat the diseases caused by AMR pathogens in animals.

Good Luck!!!

Some words or sentences should be corrected.

Reviewer 3 Report

The article on title “Occurrence of antimicrobial resistance in canine and feline bacterial pathogens in Germany under the impact of the TÄHAV amendment in 2018” by Marianne Moerer et al.,   lays out a very interesting connection between occurrence of antimicrobial resistance during 2015-2021 period for selected bacterial pathogens isolated from dogs and cats in Germany .  The manuscript is well performed and easy to follow. I think that this provides important results about the knowledge and futures perspectives for antimicrobial use  for German veterinarians. I recommend for acceptance with comments addressed below: 

1.-Abstract is adequately described.

2.- The introduction provide sufficient background and include relevant references. However, information about the impact of the TÄHAV amendment in 2018 to 2022 is necessary to understand  in terms of antimicrobial uses.

3.- The methodology is adequately described. However, how the impact of TÄHAV was considered during the study. Therefore, a major discussion is necessary with these analysis and the impact of TÄHAV, also differences between bacterial resistances from dogs and cats must to be discussed.  

Author Response

Comments and Suggestions for Authors:

We thank the reviewers for their intensive review and valuable comments

Suggestions for Authors

The article on title “Occurrence of antimicrobial resistance in canine and feline bacterial pathogens in Germany under the impact of the TÄHAV amendment in 2018” by Marianne Moerer et al.,   lays out a very interesting connection between occurrence of antimicrobial resistance during 2015-2021 period for selected bacterial pathogens isolated from dogs and cats in Germany .  The manuscript is well performed and easy to follow. I think that this provides important results about the knowledge and futures perspectives for antimicrobial use  for German veterinarians. I recommend for acceptance with comments addressed below: 

1.-Abstract is adequately described.

2.- The introduction provide sufficient background and include relevant references. However, information about the impact of the TÄHAV amendment in 2018 to 2022 is necessary to understand  in terms of antimicrobial uses.

We fully agree with you that the TÄHAV amendment wants to influence the use of antibiotics (especially HPCIA) and thus try to influence the development of resistance. Although we found in a previous survey that veterinarians perceived their use of HPCIA to be reduced (ll. 84 - 86) since the amendment, no definitive valid statement can be made until 2026 when the mandatory use rate notification will come into force (ll. 274 - 277).

3.- The methodology is adequately described. However, how the impact of TÄHAV was considered during the study. Therefore, a major discussion is necessary with these analysis and the impact of TÄHAV, also differences between bacterial resistances from dogs and cats must to be discussed. 

During this study, the TÄHAV was considered as a possible influencing factor on the development of resistance, as the amendment requires AST when a 3rd/4th generation fluoroquinolone or cephalosporin is to be used and thus had an impact on the use of antibiotics (ll. 267 - 277), as well as on testing behavior (ll. 278 - 281).

We decided not to perform a separate evaluation of resistance in dogs and cats because the number of isolates was insufficient for separate analysis. Since the applied breaktpoints were identical, it was decided to perform a joint evaluation.

Reviewer 4 Report

The article describes analysis of the data on antimicrobial resistance gathered from small animal practices in the period from 2015 to 2021. Analyzed were eight bacterial species isolated from dogs and cats and their antimicrobial profile. The article is interesting and valuable in this field of research. Important findings include increased samples submissions in Germany (encouraged by the legislative amendment in 2018), maintenance of the efficacy of antimicrobials against selected pathogens and reduction in the occurrence of resistance in several cases.

The article is adequately organized, with appropriate introduction, description of methodology and results. Discussion is well written and referenced.

There is only one remark and it is for Conclusion. The authors should consider adding additional sentence that shortly summarize this important findings on resistance trends.

Therefore I recommend it for publishing after this minor revision.

Author Response

Comments and Suggestions for Authors

The article describes analysis of the data on antimicrobial resistance gathered from small animal practices in the period from 2015 to 2021. Analyzed were eight bacterial species isolated from dogs and cats and their antimicrobial profile. The article is interesting and valuable in this field of research. Important findings include increased samples submissions in Germany (encouraged by the legislative amendment in 2018), maintenance of the efficacy of antimicrobials against selected pathogens and reduction in the occurrence of resistance in several cases.

The article is adequately organized, with appropriate introduction, description of methodology and results. Discussion is well written and referenced.

There is only one remark and it is for Conclusion. The authors should consider adding additional sentence that shortly summarize this important findings on resistance trends.

The most important finding of this study in regards on resistance trends was that for all the bacterial species, the efficacy of analyzed substances was maintained over the study period (ll. 471 - 472).

Round 2

Reviewer 1 Report

The paper has been improved and I would accept it for publication. 

Reviewer 2 Report

Thank you for your reply.

This is my last suggestion:

Lines 27-28, 113-116: The writing style of bacterial names should be changed.

For example,

Staphylococcus (S.) pseudintermedius” should be changed to “Staphylococcus pseudintermedius (S. pseudintermedius)

Escherichia (E.) coli” should be changed to “Escherichia coli (E. coli)

Pasteurella (P.) multocida” should be changed to “Pasteurella multocida (P. multocida)

Staphylococcus (S.) felis” should be changed to “Staphylococcus felis (S. felis)

Line 305, 358, 441: check spp.

Line 425: The type, model, and manufacturer of MALTI TOF should be provided.

Line 441: remove (K.)

I have no further comments.

Good Luck!!! 

I have no further comments.

Reviewer 3 Report

The authors have made a great job and sufficiently improved the manuscript. Most of my comments were addressed. I recommend for acceptance.